# Mechanism Study and Performance Evaluation of Nano-Materials Used to Improve Wellbore Stability

Yan Ye [1,2], Hanxuan Song [2,3,*], Jinzhi Zhu [4], Weiru Zheng [1,2], Fujian Zhou [2,3], Guangxu Zhou [1,2] and Qingwen Zhang [1,2]

1 School of Petroleum Engineering, China University of Petroleum, Beijing 102249, China
2 State Key Laboratory of Petroleum Resource and Prospecting, China University of Petroleum, Beijing 102249, China
3 Unconventional Petroleum Research Institute, China University of Petroleum, Beijing 102249, China
4 Oil and Gas Engineering Research Institute of Tarim Oilfield, Tarim 843300, China
* Correspondence: 2019215169@student.cup.edu.cn

**Abstract:** In the drilling process of Tarim Oilfield, a representative of ultra-deep oil and gas reservoirs, there are many problems of wellbore stability/instability caused by the development of a large number of micro-fractures. According to the nano-plugging mechanism, rigid nano-$SiO_2$ and deformable nano-paraffin emulsion are added to the drilling fluid to improve the plugging rate. The effect of nanomaterials on the mechanical properties of limestone in the Karatal Formation was evaluated through a triaxial mechanical experiment, and it was found that rigid nano-$SiO_2$ can have a better plugging effect under high formation pressure. It can increase the compressive strength of the limestone core by 10.32% and the cohesion of the core by 12.19%, and the internal friction angle of the core was increased from 25.67° to 26.39°. The internal structure of the core after nano-blocking was observed using CT scanning, and the fracture distribution state of the core before and after plugging and the fracture characteristics of the core under the pressure gradient were obtained, which confirmed that nano-$SiO_2$ can effectively solve the fracture problem of deep limestone caused by micro-fractures.

**Keywords:** rock mechanics; nanomaterials; CT scan; Tarim Oilfield





## 1. Introduction

Tarim Oilfield [1,2] is rich in oil and natural gas reserves and plays a pivotal role in China's energy reserves [3,4]. In recent years, with the deepening of the strata encountered, the drilling of ultra-deep wells [5] has become a major and difficult technical problem [6]. In the ultra-deep well in the YingSha Block of Tarim Oilfield [7,8], due to the high stress of the overlying stratum, the rock is compacted and the structure is tight; micro-fractures and micro-cavities are relatively developed [9], which leads to the problem of wellbore instability during the drilling process [10,11].

From the perspective of rock mechanics [12–14], the development of micro-fractures will destroy the integrity of the rock, weaken the mechanical properties of the original rock, and provide a channel for drilling fluid to enter the formation during the drilling process [13,15]. Under the action of drilling positive pressure difference and capillary force, the working fluid filtrate invades the formation along fractures or micro-fractures [16]. On the one hand, hydraulic fracturing may be induced, which will aggravate the rock breakage of the wellbore formation, and on the other hand, the drilling fluid and the clay in the formation will be increased [17–19], as will the probability and degree of interaction of minerals and organic matter [10,20,21].

For the wellbore instability problem of ultra-deep wells, an effective measure is to improve the plugging performance of the drilling fluid [19,22]. Due to their special scale effect, nanomaterials are used as drilling fluid additives to form dense mud cakes and

block micro-nano pores [23–25]. Drilling fluids containing nano-plugging materials [26] and can reduce the probability of wellbore instability [27,28]. Hui Mao [29] prepared a core–shell structure nano-silica composite material, which can effectively block the formation of micro-pores in ultra-deep wells and improve the bearing capacity of the formation. Song Hanxuan [30] prepared a dispersion emulsion containing a rigid core, which was successfully applied to the ultra-deep well in southwest Tarim, effectively solving the wellbore instability problem in this block. Lan Ma [31] used modified carbon nanotubes for plugging, which improved the plugging efficiency by about 50%. Nano-plugging technology has been successfully and widely used, but scholars have not researched the impact of nano-plugging on rock mechanical properties [32,33].

During the drilling process, examining the characteristics of rock mechanics changes can effectively guide drilling and avoid well wall destabilization accidents. Haiyan Zhu [34] established a prediction model of rock mechanics parameters based on the results of uniaxial compressive strength (UCS), drillability, hardness, plasticity, and abrasiveness tests of cores from complex formations to improve the drilling rate in the western Nanhai oilfield. This was successfully applied in the Nanhai block, effectively improving the speed of the block. Dingyi Hao [33] took bituminous coal as their research object and studied the development law of coal fracture structure under the action of uniaxial compression, tension, and shear. Additionally, the three-dimensional reconstruction of coal and its fractures with CT scanning presents the spatial distribution of the coal matrix, minerals, and fractures under different loading conditions in a visual and three-dimensional way. Li Huamin [12] studied the composition, microstructure, and pore distribution characteristics of rocks using an X-ray diffractometer, scanning electron microscope observation, and low-field nuclear magnetic resonance, and conducted uniaxial compression tests on rocks to obtain the impact of the relationship between composition, microstructure, and pore characteristics on rock mechanics. Houbin Liu [14] experimentally tested the physical, chemical, and mechanical properties of shales from the Longmaxi Formation immersed in different working fluids. Combining with the test data, a theoretical model was developed to evaluate and analyze the wellbore stability of the Longmaxi Formation shale under different working conditions of drilling, fracturing, and completion. Jianhua He [35] studied the mechanical properties of shale reservoirs and their influencing factors through triaxial high-temperature and high-pressure tests, fracture toughness, and X-ray diffraction experiments, taking the deep marine shale reservoir of the Wufeng-Longmaxi Formation in southern Sichuan as an example, and studied the rock fracture morphology quantified under various loading conditions. Based on the morphological characteristics of shale, the analysis of influencing factors and comprehensive quantitative evaluation of shale brittleness were carried out. The interpretation of well wall stability during drilling by rock mechanics testing is a current focus area and direction of research in the drilling field.

In this paper, triaxial compression and CT scanners are used to carry out related research [9,36]. According to the investigation of the drilling situation in the southwestern area of Tarim, it is found that the problem of broken limestone cores in the Karatal Formation has seriously affected the exploration and development process here [37]. The addition of nanoparticles in water-based drilling fluids can effectively block rock micro-pores, maintain wellbore stability, and improve rock mechanical strength [10,15]. In this paper, we investigate the mechanism of nanomaterials to improve rock mechanics.

## 2. Materials and Methods

### 2.1. Materials

The nano-silica comes from the modified dispersed nano-silica prepared by the China University of Petroleum (Beijing), and the paraffin microemulsion comes from the paraffin micro-fluid prepared by the China University of Petroleum (Beijing).

The cores were taken from the limestone outcrop of the Karatal Formation (the well is located at a depth of 6260 m, and the main lithology is grey-green mudstone and cloudy tuff) in the southwestern part of the Tarim Basin. During the drilling process, drilling fluid

leakage can cause microfracture development, which can lead to well wall instability. The sampling results are shown in Figure 1.

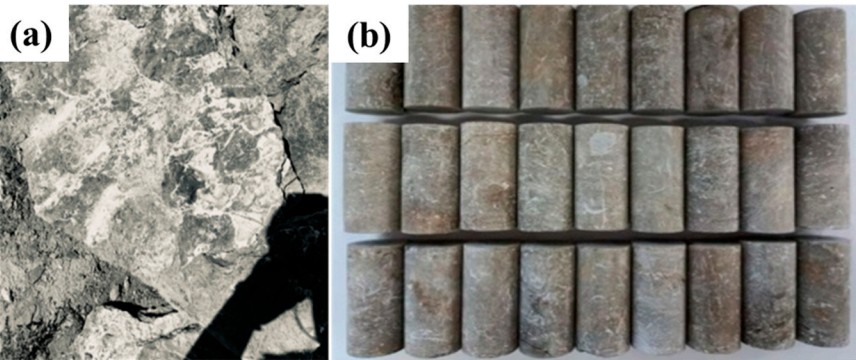

**Figure 1.** (**a**) Geological outcrop of Karatal Formation; (**b**) Drilling core of Alatal Formation.

*2.2. Method*

2.2.1. Transmission Electron Microscope Testing

This test is used to study the microscopic properties and structure of nanomaterials. The microstructure was observed using transmission electron microscopy (TEM). The paraffin microemulsion and the modified nano-silica solution were dropped into the metal copper mesh, left to dry, and then placed on the stage for observation.

2.2.2. Drilling Fluid Performance Test

In this test, a six-speed viscometer is used to measure the rheological properties of the drilling fluid, and a high-temperature and high-pressure fluid loss meter are used to measure the fluid loss additives of the drilling fluid. The above experiments were used to analyze the effect of additives on the basic performance of drilling fluids.

2.2.3. Scanning Electron Microscope Testing

The core samples as well as the drilling fluid filter-off cake were dried in a vacuum drying chamber at 60 °C. Before analysis, the sample was attached to an ion sputtering device for gold spraying to improve imaging quality and then observed.

2.2.4. Particle Size Analysis Testing

A certain agglomeration effect occurs when nanomaterials are dispersed in an aqueous solution, and the particle size distribution of the dispersions was measured using a Mastersizer 3000 laser diffraction particle size analyzer (Mastersizer, Worcestershire UK).

2.2.5. Rock Triaxial Mechanical Testing

This experiment uses two sets of equipment: a high-temperature and high-pressure dynamic soaking-drive device and rock mechanical parameter test equipment (Axial pressure:1000 kN; Surrounding pressure:140 MPa; T:150 °C). The experimental device of the rock three-axis stress test system is a set of rock ground stress comprehensive test systems produced by the company US GCTS.

Different formulations of drilling fluids were used to soak the cores in the high-temperature and high-pressure dynamic soaking-drive device with heating and pressure for a certain length of time. The soaked cores were made into samples for rock mechanics experiments.

The main system module of the RTR-1000 triple-axis rock mechanical experimental machine consists of, a stress loading system, enclosure loading system, axial and radial deformation measurement system, servo control and data acquisition system, ultrasonic and acoustic transmission measurement system, radial speed testing system and differences in the straight test system. It can test and analyze the sound wave rate, geographical stress,

and mechanical strength of rock hearts under high-temperature and high-pressure strata. The experimental device is shown in Figure 2.

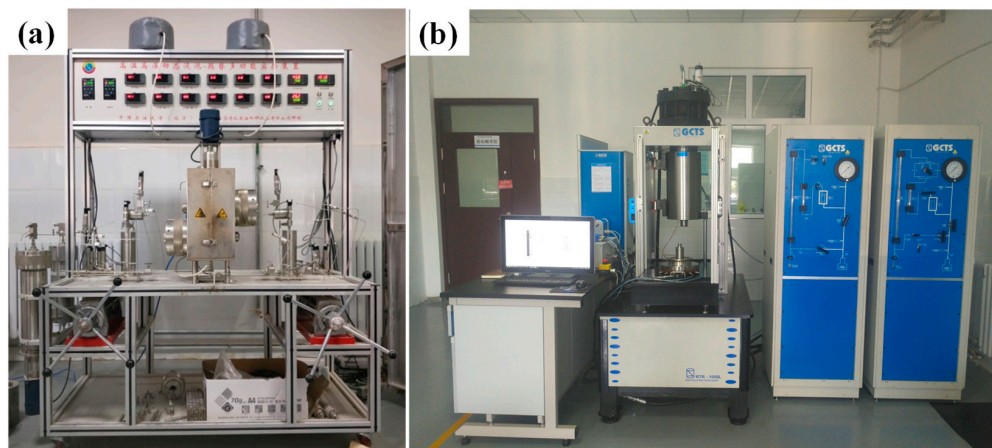

**Figure 2.** (**a**) High-temperature and high-pressure dynamic immersion-displacement device (**b**) RTR-1000 triaxial rock mechanics experimental machine.

The main objectives of triaxial tests are to determine (1) rock damage parameters, mainly Mohr–Coulomb cohesion and friction angle under representative reservoir loading/stress conditions; (2) uniaxial compressive strength (UCS) extrapolated from damage strength under confinement; (3) static modulus of elasticity, Young's modulus and Poisson's ratio under representative reservoir loading/stress conditions

2.2.6. CT Core Scanning Experiment

The limestone of the Karatal Formation was soaked in drilling fluid before and after optimization was used, and then the medical CT scanning experiment was carried out on the soaked samples [38]. Then, Avizo software was used for image processing to analyze the plugging effect on the fracture path after the test. The CT scanner used in the experiment is shown in Figure 3.

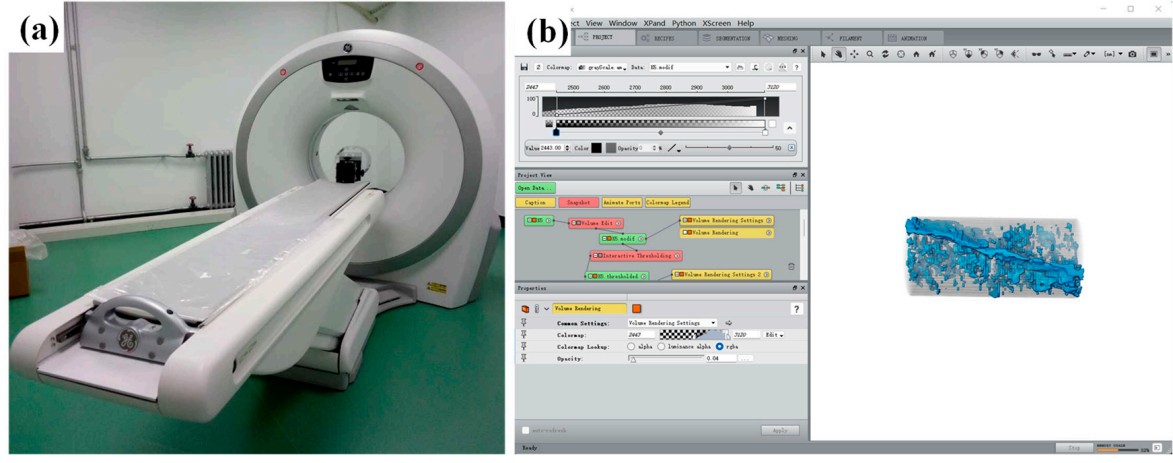

**Figure 3.** (**a**) Medical CT Scanner; (**b**) Avizo software interface.

## 3. Results and Discussion

### 3.1. Performance Analysis of Nanomaterials for Optimizing Drilling Fluids

3.1.1. Micromorphology and Particle Size Analysis of Nanomaterials in Solution State

In this paper, we analyze the mechanism of nanomaterials to maintain well wall stability from the perspective of rock mechanics, and firstly, we investigate the properties of commonly used nanomaterials. Paraffin microemulsion and modified nano-$SiO_2$ are

added to the potassium-based polysulfone drilling fluid applied in the field to expand the plugging gradation, form a broad-spectrum plug, plug micro-nano pores, and fractures, and effectively solve the problem of micro-fracture development caused by the wellbore instability problem. The micrographs and particle sizes of paraffin microemulsion and modified nano-$SiO_2$ are shown in Figure 4.

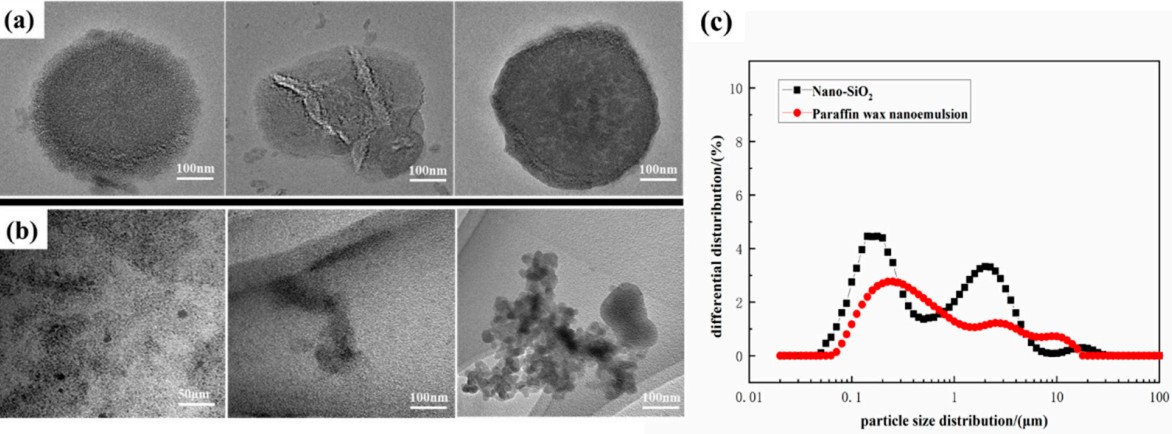

**Figure 4.** (**a**) TEM image of paraffin microemulsion; (**b**) TEM image of nano-$SiO_2$; (**c**) Particle size distribution of aqueous solutions of nanomaterials.

### 3.1.2. Analysis of Drilling Fluid Performance after Adding Nanomaterials

The nanomaterials are added to the potassium-based polysulfone drilling fluid used on-site to form the corresponding three sets of technical formulas:

(0) Potassium-based polysulfonic drilling fluid: 3%bentonite + 0.4%NaOH + 0.3% PAC + 0.1% AP-220 + 4% SMP-3 + 4% SPNH + 4% Oxidized Asphalt + 3% KCl+ barite

(1): 0# + 0.5% paraffin microemulsion

(2): 0# + 0.8% nano-$SiO_2$

The rheological fluid loss properties of drilling fluids with different formulations are shown in Table 1.

**Table 1.** Rheological fluid loss performance of drilling fluid.

| Number | Density g/cm$^3$ | Temperature/Time | AV/MPa·s | PV/MPa·s | Yp/Pa | G10′/G10″ | HTHP Filtrate Volume | The Thickness of Filter Cake |
|---|---|---|---|---|---|---|---|---|
| 0 | 1.95 | 150 °C/16 h | 47 | 40 | 7 | 2.5/16 | 7.4 | 2.0 |
| 1 | 1.95 | 150 °C/16 h | 44 | 38 | 6 | 2/18 | 5.2 | 2.1 |
| 2 | 1.95 | 150 °C/16 h | 38 | 32 | 6 | 2.5/17.5 | 3.6 | 1.2 |

After adding nanomaterials, the drilling fluid has a good streaming performance, and the viscosity is slightly reduced. Among them, the high temperature and high-pressure dehydration are reduced from 7.4 mL to 3.6–5.2 mL. The thickness of the clay cake was reduced from 2.0 mm to 1.2 mm, which greatly improved the water loss of the drilling fluid.

### 3.2. Core Microstructure Analysis

During the drilling process, we obtained the crushing strata core in the southwest YS area of Tarim. It can be seen from the field data that the fractured zone is mainly from the limestone of the Karatal Formation. It has a large number of micro-nano pores and cracks inside. The micro-photo of the core is shown in Figure 5.

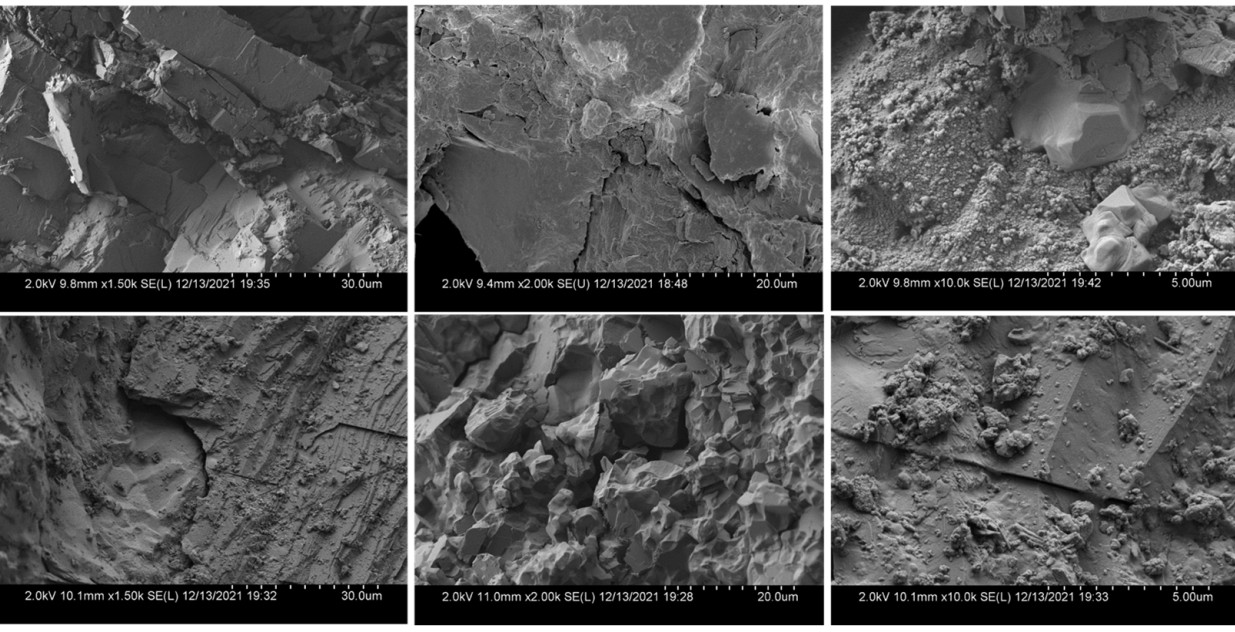

**Figure 5.** SEM test of cores from Karatal Formation.

According to the Karatal group's core SEM photo, it can be seen that the group of rocks is formed by a large number of particles and chroma that are not closely piled up, accompanied by the development and accumulation of pores produced with micro-nano-cracks. The opening of the fracture is between 10–300 nm, which easily causes leakage of drilling fluid, reduces rock strength, and causes well wall destabilization.

### 3.3. Rock Triaxial Mechanical Analysis

3.3.1. Calculation of Rock Mechanics Parameters

The triaxial compressive strength (CCS), elastic modulus, and Poisson's ratio of rocks are usually obtained from the data obtained between elastic deformations to determine Young's modulus E and Poisson's ratio $v$ for each restraint stress stagnation point. The stress–strain relationship must be in the linear elastic range.

Unconfined compressive strength (UCS) is calculated using Formula (1), Young's modulus E is calculated using Formula (2), and Poisson's ratio is calculated using Formula (3).

The UCS intensity is determined by the ratio of the load L to the cross-sectional area A of the sample,

$$UCS = \frac{L}{A} \tag{1}$$

Young's modulus E is determined by the slope of the axial stress ($\sigma_a$) versus axial strain ($\varepsilon_a$) curve,

$$E = \frac{\sigma_a}{\varepsilon_a} \tag{2}$$

Poisson's ratio is determined by the slope of the radial strain ($\varepsilon r$) versus axial strain curve,

$$v = -\frac{\varepsilon_r}{\varepsilon_a} \tag{3}$$

*L: load*
*A*: sample cross-sectional area
$\varepsilon r$: radial strain
$\sigma a$: axial stress
$\varepsilon a$: axial strain

The Mohr–Coulomb theory mathematically describes the response relationship between normal stress $\sigma$ and shear stress $\tau$ in brittle materials and is applicable to materials

whose compressive strength far exceeds tensile strength. A Mohr circle is a two-dimensional graphical representation of the state of shear stress and normal stress at any point, and the circumference of the circle is the locus of points representing the shear state and normal stress on a single plane. The Moiré failure envelope is the tangent of the largest possible circle obtained at different stresses, and the rock will fail when it reaches the stress state in the portion above this tangent. Combining the Mohr failure criterion with the Coulomb equation yields straight lines for most Mohr circles, and the Mohr–Coulomb failure envelope relationship is defined by Equation (4).

$$\tau = c + \sigma \tan\theta \tag{4}$$

where c is the cohesion and $\theta$ is the angle of internal friction. The cohesive strength is defined as the intrinsic shear strength or cohesion of the material.

The Mohr–Coulomb damage model considers only the maximum stress $\sigma_1$ and the minimum principal stress $\sigma_3$, and the intermediate principal stress $\sigma_2$ does not play a role. Usually acting on the rock, the rock is stable when the deviatoric stress, represented by a Mohr circle of diameter $\sigma_1$–$\sigma_3$, is kept below the damage envelope. If the deviatoric stress is high enough that the circle crosses the damage envelope, the rock will fail. Beyond the damage envelope is the region of plastic deformation.

The effective confining stress $\sigma_3$ and the effective peak breaking stress $\sigma_1$ are collated for each sample. Moiré circle diagrams are then constructed by drawing circles for each sample having diameters $\sigma_1$–$\sigma_3$. Tangent lines are then drawn to connect each circle and extend to an intercept on the shear stress axis to obtain the cohesion, with the slope of the tangent line being $\tan\theta$.

The Mohr–Coulomb damage criterion can also be expressed as a linear function of the principal stresses ($\sigma_1$ and $\sigma_3$) as follows:

$$\sigma_1 = \sigma_0 + \sigma_3 k$$

where $\sigma_1$ is the maximum principal stress, i.e., the axial stress at which the rock in the experiment fails, and $\sigma_3$ is the enclosing pressure. The $\sigma_1$-$\sigma_3$ pairs from each sample are plotted on the so-called principal stress axis diagram. The intercept on the $\sigma_1$ axis is equal to the uniaxial compressive strength $\sigma_0$ at a value corresponding to or close to the value determined by the UCS test. The core strength was analyzed by analyzing the microstructure of rocks combined with the rock triaxial compressor test. Accordingly, the method of applying nanomaterials to drilling fluids to improve the mechanical properties of rocks is proposed, which provides a theoretical guidance basis for rock fragmentation and block dropping triggered by microfracture development in Tarim ultra-deep well formations.

### 3.3.2. Analysis of Mechanical Properties of Rock

Different drilling fluid formulations were used to soak the Karatal Formation limestone cores at 170 °C/3.5 MPa to obtain cores with different numbers. The stress–strain curves of the cores were tested under the confining pressure of 10 MPa/20 MPa/30 MPa, respectively. The conditions are shown in Table 2.

The mechanical properties of cores immersed in different formulations of drilling fluids vary greatly. The stress–strain curves of the cores using the drilling dip incorporating nanomaterials are closest to the original rock, and the compressive strength of the rock has been enhanced. The mechanical properties of the core calculated from the stress–strain curve are shown in Table 3.

**Table 2.** Rock mechanics experimental parameters.

| Source | Number. | Core Number | Experimental Conditions | Confining Pressure (MPa) |
|---|---|---|---|---|
| Karatal Formation limestone | a | a-1<br>a-2<br>a-3 | - | 10<br>20<br>30 |
| | b | b-1<br>b-2<br>b-3 | No.0 drilling fluid soaking | 10<br>20<br>30 |
| | c | c-1<br>c-2<br>c-3 | No.1 drilling fluid soaking | 10<br>20<br>30 |
| | d | d-1<br>d-2<br>d-3 | No.2 drilling fluid soaking | 10<br>20<br>30 |

**Table 3.** Triaxial mechanical test of Karatal Formation Core before and after soaking.

| Source | Core Number | Compressive Strength (Principal Stress) (MPa) | Elastic Modulus (GPa) | Poisson's Ratio | Cohesion (MPa) | Internal Friction Angle (°) |
|---|---|---|---|---|---|---|
| Karatal Formation limestone | a-10 | 135.327 | 26.813 | 0.255 | | |
| | a-20 | 166.020 | 24.756 | 0.268 | 34.890 | 25.920 |
| | a-30 | 186.395 | 24.741 | 0.272 | | |
| | b-10 | 120.256 | 23.653 | 0.265 | | |
| | b-20 | 151.398 | 24.717 | 0.273 | 30.480 | 25.670 |
| | b-30 | 170.823 | 26.463 | 0.261 | | |
| | c-10 | 125.096 | 26.033 | 0.321 | | |
| | c-20 | 130.454 | 19.650 | 0.318 | 30.590 | 24.350 |
| | c-30 | 173.153 | 21.019 | 0.303 | | |
| | d-10 | 134.223 | 24.649 | 0.257 | | |
| | d-20 | 174.872 | 22.055 | 0.238 | 34.730 | 26.390 |
| | d-30 | 187.312 | 19.480 | 0.272 | | |

Considering the influence of rock heterogeneity, the experimental results such as compressive strength and elastic modulus will fluctuate within a certain range. To better show the experimental results, the cohesion calculation is usually used to characterize the rock strength, which is more representative. The mechanical parameters of the unsoaked protolith were counted as 100%, and the rock mechanical test data under different soaking conditions are shown in Table 4. The comparative analysis of compressive strength, elastic modulus, and cohesion under different soaking conditions is shown in Figure 6.

**Table 4.** Rock mechanical strength of Karatal Formation core before and after soaking.

| Source | Number | Compressive Strength (MPa) | Elastic Modulus (GPa) | Cohesion (MPa) |
|---|---|---|---|---|
| Karatal Formation limestone | a | 100.00% | 100.00% | 100.00% |
| | b | 88.86% | 88.21% | 87.35% |
| | c | 92.44% | 97.09% | 87.68% |
| | d | 99.18% | 91.93% | 99.54% |

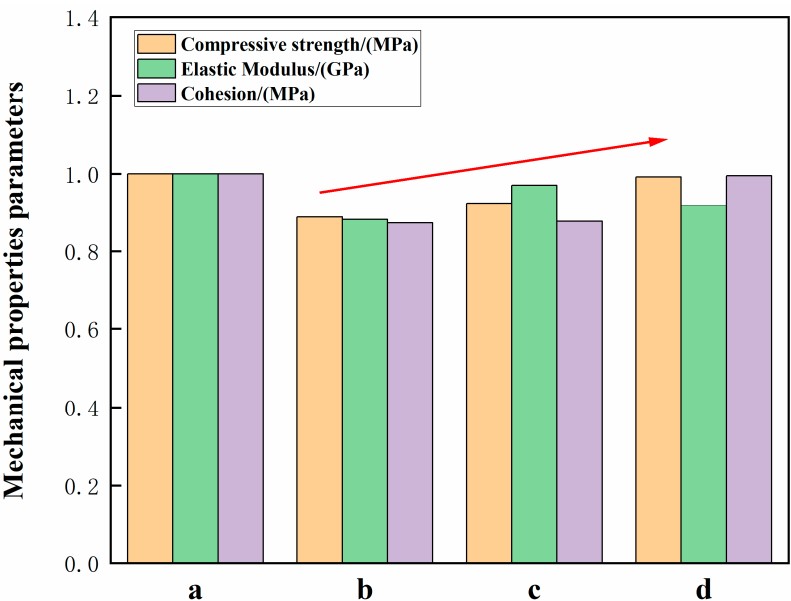

**Figure 6.** Comparison of rock mechanics strength of Karatal Formation Core before and after soaking.

It can be seen from Figure 6 that for the limestone of the Karatal Formation, after soaking in No.0 drilling fluid, the compressive strength of the rock is reduced to 88.86% of the protolith, the elastic modulus is reduced to 88.21% of the protolith, and the cohesion is reduced to 87.35% of the protolith. After soaking in the No. 1 drilling fluid with the addition of paraffin microemulsion, the compressive strength of the rock was reduced to 92.44% of the protolith, the elastic modulus was reduced to 97.09% of the protolith, and the cohesion force was reduced to 87.68% of the protolith. After soaking in No. 2 drilling fluid with nano-$SiO_2$, the compressive strength of the rock was reduced to 99.18% of the protolith, the elastic modulus was reduced to 91.93% of the protolith, and the cohesion was reduced to 99.54% of the protolith. The addition of nano-$SiO_2$ as a plugging agent can better retain the mechanical strength of the rock. This is because the nano-$SiO_2$ has better compressive performance and can be well cemented with the rock to form a plug body [39]. The analysis of the rock force parameters test is a good reflection of the effect of drilling fluid on rock mechanical strength. Therefore, the rock mechanical strength test can also be used as one of the evaluation methods for drilling fluid plugging additives.

*3.4. CT Microstructural Analysis*

According to the calculation results of the experimental results, the mechanical properties of the cores are different under different formulations of drilling fluid immersion. CT scanning was used to further explore its fracturing fracture morphology. The CT scan results were reconstructed in image processing software (Avizo) to obtain the internal micro-morphology of the core, as shown in Figure 7.

It can be seen from Figure 7a that the rock strength of the protolith is high, the rock is densely cemented, and no micro-fractures develop when no drilling fluid invades; Figure 7b is the core soaked in the field drilling fluid. Therefore, the filtrate intrudes into the core along the micro-nano pores, causing damage to the internal structure of the core and secondary development of micro-fractures, reducing the mechanical strength of the rock. In Figure 7c,d, adding nanomaterials to the drilling fluid effectively plugs the micro-nano pores, which are similar to the protolith fracture morphology, confirming the plugging performance of the nanomaterials, which can effectively improve the mechanical strength of the rock. When the nanomaterials enter the cracks, they can effectively seal the cracks and reduce the further invasion of filtrate, which causes the degradation of rock mechanical properties. Additionally, $SiO_2$ has a better supporting property, which improves the pressure resistance of the rock to a certain extent. It is proven that nanomaterials can effectively improve well

wall stability. CT scanning can be used to effectively observe the core fracture morphology and achieve a visual and quantitative description of the damage caused by drilling fluid to the core, and in this study, the effect of nanoadditives on the rock pores as well as fracture development can be observed. However, considering the non-homogeneity of the fractures, a combination of evaluation methods is needed to achieve an accurate portrayal of the mechanical properties of the core.

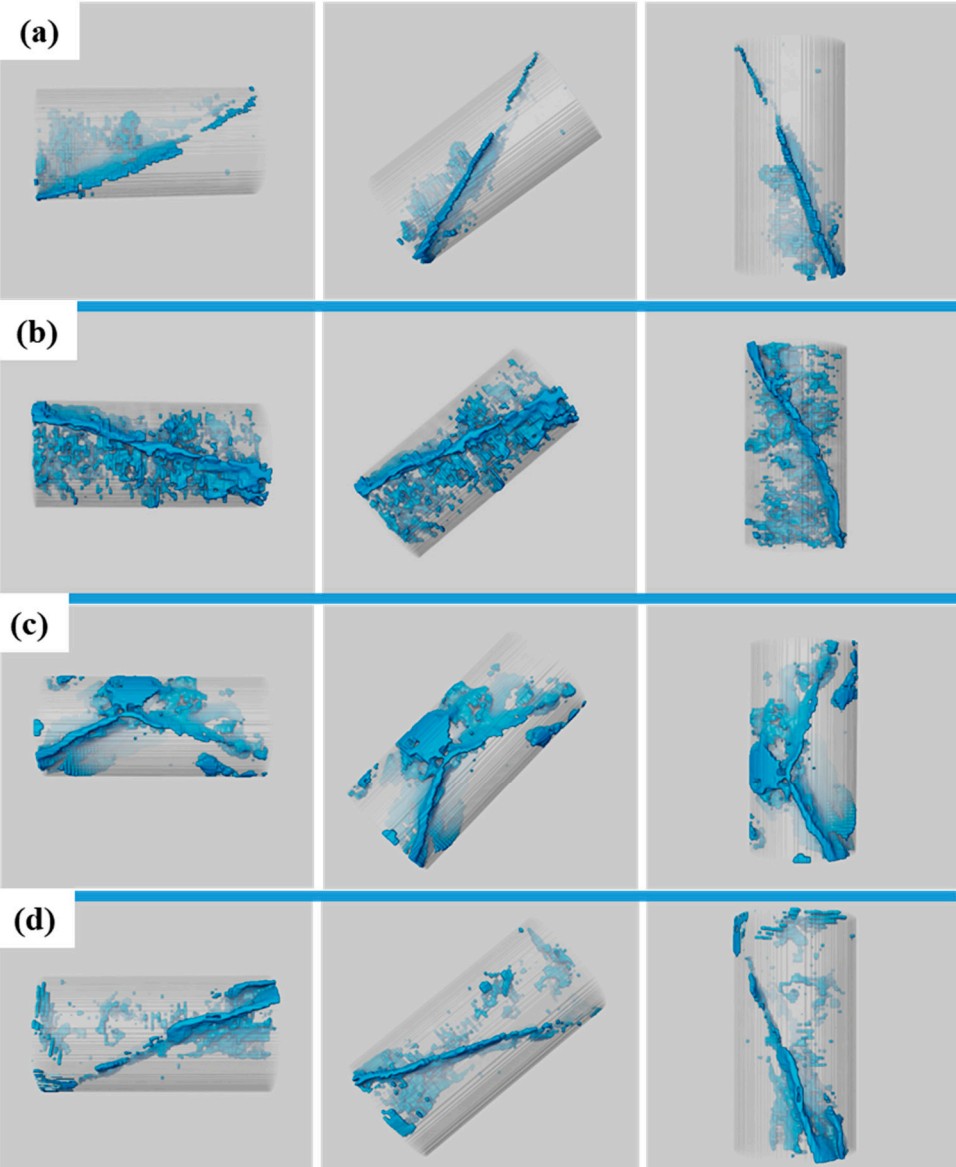

**Figure 7.** (**a**) Forms of fracturing fractures in protolith; (**b**) Forms of fracturing fractures in cores soaked in drilling fluid on site; (**c**) Forms of fracturing fractures in cores soaked in paraffin microemulsion drilling fluid; (**d**) Fractures in cores soaked in nano-$SiO_2$ drilling fluid form.

## 4. Nanomaterials Reduce the Mechanism of Rock Mechanical Strength Change

When the drilling fluid enters the formation, due to the large particles in the drilling fluid, it cannot seal the micro-nano pores of the formation, causing the filtrate in the drilling fluid to enter the formation and further react with the rock minerals, which will easily cause the further development of fractures. For hard and brittle tuffs, the fractures are prone to breakage and block dropping, resulting in block dropping and stuck drilling accidents. When nanomaterials are added to the drilling fluid, the probability of fracture development is reduced and a broad-spectrum sealing effect is achieved. When there are fewer cracks in

the rock, the rock has better compressive strength and cohesion under the effect of multiple stresses. The mechanism of action is shown in Figure 8.

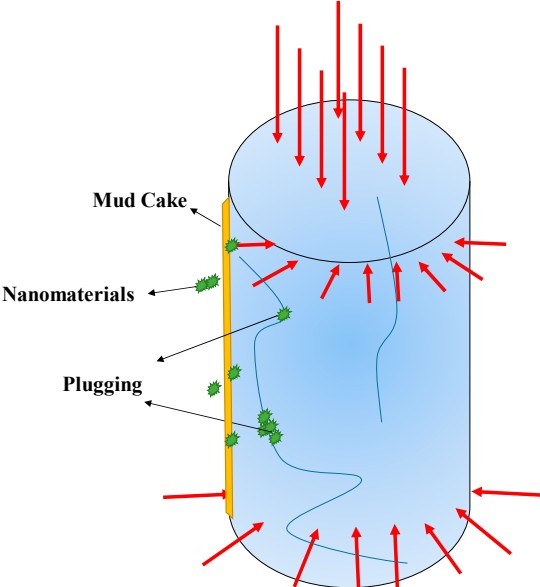

**Figure 8.** Mechanism of action of nanomaterials.

## 5. Conclusions

The fractured strata rocks in the southwestern area of Tarim are mainly limestones of the Karatal Formation, which contain a large number of micro- and nano-fractures and pores. On-site drilling fluid immersion will lead to the development of micro-fractures in the rock and reduce the mechanical strength, causing wellbore instability.

1. For micro-fractures in high-stress formations, the plugging efficiency of rigid nano-$SiO_2$ is higher than that of deformed plugging material nano-paraffin emulsion.

2. Nano-$SiO_2$ can increase the compressive strength of the limestone core by 10.32% and the cohesion of the core by 12.19%, increasing the internal friction angle of the core to 26.39°.

3. The dispersed nano-$SiO_2$ plugging agent can form a plugging body in the near-wellbore zone, reducing the damage and fragmentation of the filtrate to the core.

4. Nano-blocking can reduce the infiltration of filtrate along micro-fractures and prevent the secondary development of natural micro-fractures.

This study shows that nanomaterials can reduce the damage to the reservoir and effectively retain the rock strength when used as drilling fluid additives, which also provides new ideas for ultra-deep well drilling fluid optimization.

**Author Contributions:** Y.Y.: Funding acquisition, Project administration. H.S.: Conceptualization, Methodology, Writing—original draft. J.Z.: Funding acquisition, Project administration. W.Z.: Investigation, Formal analysis. F.Z., G.Z. and Q.Z.: Writing—review and editing. All authors have read and agreed to the published version of the manuscript.

**Funding:** This research received no external funding.

**Institutional Review Board Statement:** Not applicable.

**Informed Consent Statement:** Not applicable.

**Data Availability Statement:** Not applicable.

**Conflicts of Interest:** The authors declare no conflict of interest.

## Nomenclature

| | |
|---|---|
| AV | Apparent viscosity |
| PV | Plastic viscosity |
| Yp | Yield point |
| G10′ | Initial gel strength |
| G10″ | 10 min gel strength |
| HTHP | High-temperature and high-pressure |
| SEM | Scanning electron microscope |
| TEM | Transmission electron microscope |

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
