# Peer review of "Mechanism Study and Performance Evaluation of Nano-Materials Used to Improve Wellbore Stability"

_sustainability, doi:10.3390/su15065530_

Round 1

Reviewer 1 Report

The manuscript entitled "Nano-plugging materials in drilling fluids can solve the problem of mechanical fracture of ultra-deep limestone" was read and reviewed in details. In this study, the effect of nanomaterials on the mechanical properties of limestone in the Karatal Formation was evaluated by a triaxial mechanical experiment, and it was found that rigid nano-SiO2 can play a better plugging effect under high formation pressure. This manuscript is in the scope of the journal of sustainability and can be accepted after major revision. Therefore, I suggest the following modifications:

Major comments:

1- The introduction can be improved by considering some relevant studies in hydraulic fracturing and mechanical behavior of clay rich rocks as follow:

1-     "Experimental observations on the effect of strain rate on rock tensile fracturing." International Journal of Rock Mechanics and Mining Sciences 160 (2022): 105256.

2-     "Comprehensive investigation of rock fracture behavior in clay-rich rocks under the effect of temperature: Experimental study under three loading modes (I, I/II, II)." Engineering Fracture Mechanics 276 (2022): 108933.

2- How was strain and displacement measured in this study? To determine the Young's modulus and Poisson's ratio accurately, vertical and lateral strain of the rock sample should be accurately measured, this can be achieved by using special techniques such as strain gauges, this is because the displacements measured by the servo-hydraulic device do not actually correspond to the real displacements occurring in the rock sample. How was this issue addressed in this work?

3- How was the cohesion and internal friction angle calculated, pleases explain in detail the used equations?

Minor comments:

1- Line 8: "wellbore stability/instability" instead of " wellbore stability instability"

2- Line 9: increasing the friction angle of the core to 26.39o, how was the friction angle before the increase?

3- Line 73: What are the maximum temperature and pressure provided using a triaxial machine (US GCTS)?

4- Line 87, it is better to say "to analyze plugging effect on fracture path after test" instead of " to analyze fracture development and plugging effect".

5- Please explain the abbreviations (AV, PV, Yp/pa, G10'/G10", HTHP) in table 1 and anywhere throughout the manuscript.

6- Third column in table 1, I think "150oC/16h" instead of "150oC*16h"

7- "Mohr circle" instead of "Moire circle" throughout the manuscript

8- Please enlarge the numbers and data mentioned in figure 6.

9- Lines 159-160: "After adding nanomaterials, the curves of groups (b) and (c) appear at multiple inflection points, 159 which are close to the formation core curves." What do you mean by this sentence?

10- Line 175: "reduced to 87.35" instead of "reduced to 88.21"

11- Line 175: "of the protolith. 87.35% of the protolith" please correct this sentence.

Author Response

Thank you very much for your valuable comments and suggestions on our manuscript. Following the reviewers’ comments, we have modified and improved our manuscript according to your kind advice and the referee’s detailed suggestions. Enclosed please find the responses to the referees. We sincerely hope this manuscript will be acceptable to be published on Sustainability.

In addition, I would like to thank the reviewers for their detailed and valuable comments.

Thank you very much for all your help and looking forward to hearing from you soon.

Best regards

Sincerely yours

Ph.D Song Hanxuan

Reviewer 2 Report

The basic mechanical tests were performed on an addictive. The work is comprehensive and quality looks well, although the depth needs improvement. The comments are listed as follows.

(1) The title should be revised to make it sound more scientific.

(2) "increasing the internal friction angle of the core to 26.39 " in the abstract: unit is lacking.

(3)  Section 2.1: It is better to show the basic properties of the rock samples and formation.

(4) No drilling blocks are found in the figure 1, while in the manuscript the blocks were stated as broken during the drilling.

(5) The details of the used addictive are lacking. how to make it and repeat the test?

(6) Could the addictives always improve the properties? For other materials, sometimes the answer is no and what is the the controlling mechanism. Maybe the discussion would be more interesting. The following references are listed which could be made in the introduction or discussions to make it attract interest.  Zhen Li, Jiachen Liu, Rongchao Xu, Huoxing Liu, Wenhao Shi. Study of grouting effectiveness based on shear strength evaluation with experimental and numerical approaches. Acta Geotech, 2021, 16: 3991-4005. 

(7) The parameters in the manuscript should be clarified, such as Eq. (3.4). Additionally, c is always used instead of So in the rock mechanics.

(8) Fig. 6 is too small, which would influence the output quality.

(9) The difference between the samples before and after the treatment from damage perspective should be focused. The damage characteristics are important for revealing the mechanism of rock failure (Plz refer to: Zhen Zhang, Zhen Li, Gang Xu, Xiaojin Gao, Qianjin Liu, Zhengjie Li, Jiachen Liu. Lateral abutment pressure distribution and evolution in wide pillars under the first mining effect. International Journal of Mining Science and Technology, 2023. https://doi.org/10.1016/j.ijmst.2022.11.006.) It is better to discuss it even if the investigation has not been conducted.

(10) What is the strength in Table 3? The deviatoric strength or the principal stress?

(11) The unit of internal friction angle is lacking in Table 3. 

(12) Fig. 7: The unit is also lacking. What are the different samples?

Author Response

(The authors gave the same response as above.)

Reviewer 3 Report

In this paper, triaxial mechanics and CT scanning methods are adopted to study the influence of nanomaterials on the mechanical properties and internal structure of ultra-deep well cores in southwest Tarim, which is innovative to a certain extent, but there are still some problems that need to be further optimized to meet the quality requirements of AD.

1.     Table 4 is missing from the article, please check.

2.     In lines 143-145, give a further description of SEM photographs of the rock.

3.     Figure 4 on line 188 is incorrect.

4.     Figure 5 on line 205 is incorrect.

5.     Figure 6 on line 227 is incorrect.

Author Response

(The authors gave the same response as above.)

Round 2

Reviewer 1 Report

No further comments

Reviewer 2 Report

Thanks for the authors' detailed works. The current version is suitable for publication.
